# Combination Weighting Integrated with TOPSIS for Landscape Performance Evaluation: A Case Study of Microlandscape from Rural Areas in Southeast China

**Lan Shen [1,2], Yikang Zhang [2], Minfeng Yao [2] and Siren Lan [1,*]**

1   College of Landscape Architecture and Art, Fujian Agriculture and Forestry University, Fuzhou 350002, China
2   School of Architecture, Huaqiao University, No. 668 Jimei Rd., Jimei Dist., Xiamen 361021, China
*   Correspondence: lansirenfjnlu@163.com; Tel.: +86-166-7852-4762

**Abstract:** This study aims to evaluate the landscape performance of rural microlandscapes in highly urbanized areas and propose optimization strategies based on the evaluation results. As a sustainable promotion mode, microlandscapes can effectively improve the damage caused by the development of rugged urbanization to the living environment. To improve the rural living environment, some achievements have been made in the construction of microlandscapes in the highly urbanized rural areas of southeast coastal areas, represented by Fujian Province, but there are still problems such as low utilization rate and difficult maintenance. As a qualitative and quantitative weighting method, the combination weighting method is widely used in the construction of evaluation models of safety engineering, environmental management, and other disciplines. This study constructed a landscape performance evaluation system based on the American landscape performance series and combined it with performance evaluation methods in other related fields to establish a landscape performance evaluation system suitable for rural microlandscapes in highly urbanized areas. Taking social benefits as an example, five main factors affecting social benefits are highlighted: comfort and health; safety and accessibility; sociability and service; aesthetics and education; and culture and inheritance. Each factor contains different sub-criteria to identify specific problems. Field observation, questionnaire survey, and interview records of 25 microlandscape projects in Yinglin Town, Jinjiang City were conducted. The combination weight calculation based on the AHP-entropy weight method and the comprehensive benefit ranking calculation based on the TOPSIS method is carried out. It was found that stress relief and the number of visitors were the main factors affecting the social benefits of microlandscape performance, and the top-ranked projects also had such characteristics. The seasonal phase and color richness had the least effect on social benefits. Therefore, the microlandscape should improve the healing effect of the project on users as much as possible in the design stage, so that users can better relax through the microlandscape. In addition, strategies such as space selection and path optimization should be adopted to improve the utilization rate of the microlandscape as much as possible, and the fairness of the use of vulnerable groups should be fully considered.

**Keywords:** rural living environment; rural microlandscape; highly urbanized areas; landscape performance evaluation; evaluation method; combination weighting method

## 1. Introduction

Rural areas are an important part of human settlements. According to the United Nations Committee on Population and Development, as of 2020, approximately 44 percent of the global population still lived in rural areas. In 2017, the Chinese government put forward a rural revitalization strategy, making solving problems in agriculture, rural areas, and farmers its top priority. In this context, an increasing number of scholars have begun to study the rural living environment from the perspectives of urban and rural planning, ecological environment science, and sociology [1–5]. Villages in the southeast coastal

areas of China mostly exhibit a high degree of urbanization. Their pillar industries are no longer agriculture but secondary and tertiary industries, such as industry and service industries. Some local enterprises in these villages attract migrant populations to work. Thus, it is important to improve the living environment of these villages. In the process of urbanization, the phenomenon of housing demolition and construction has occurred. Due to poor management and unclear rights and responsibilities in rural areas, construction waste is not cleaned up in time, and the ongoing pollution caused by enterprise production has caused serious damage to village environments. It is necessary to clean up waste, repair the polluted environment and improve the quality of the rural living environment.

The rural microlandscape is a means to improve the rural living environment with low cost and high-efficiency techniques using abandoned local materials and traditional construction and encouraging villagers to participate in construction projects. In recent years, it has become the preferred solution to improve the spatial quality and living environment of highly urbanized rural areas. However, the existing microlandscape has many problems, such as a lack of maintenance, short survival time and low utilization rate. To explore the causes of these problems, this study attempts to analyze the benefits achieved in all aspects of the completed microlandscape through field research and proposes optimization strategies to better guide the design, construction, and maintenance of new microlandscape projects.

Performance evaluation of the rural microlandscape, as a whole life cycle of the landscape, has guiding significance for most parts of landscape studies (Figure 1). It can help professional designers, researchers and managers in the analysis and design of rural landscapes to re-examine the different functions of each link. The optimization of microlandscape design for the whole life cycle, management and maintenance should be considered at the beginning of the project to improve efficiency. As the core part of the whole landscape evaluation system, the landscape performance evaluation model of the rural microlandscape is not only related to how to process data scientifically and correctly, but also determines the performance ranking of different microlandscape projects to better guide later projects. However, due to the late start of landscape performance evaluation for rural microlandscapes in China, and the fact that the evaluation systems are not suitable for microlandscape evaluation in rural areas, the existing evaluation system has various problems. Therefore, it is crucial to develop a landscape performance evaluation model that is appropriate for rural microlandscape to improve the living environment in rural areas and generate more and accurate feedback on the construction of microlandscapes.

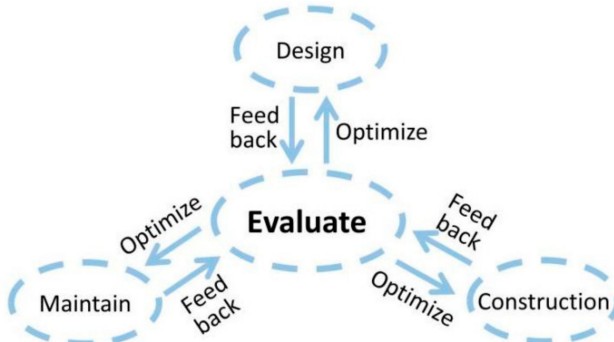

**Figure 1.** The whole life cycle process relationship of the microlandscape.

At present, the relevant research on landscape performance evaluation methods has achieved relatively mature results, but there are few specific landscape performance evaluation methods for microlandscapes in rural areas, and a systematic and standardized system has not been formed. The majority of the early published research concentrated on the analysis of urban landscape performance, and the assessment techniques employed were mostly straightforward. Zhang Sujuan et al. (2008) evaluated the social benefits of the Qinhuangdao landscape by evaluation (SBE) [6]; Gao Dongmei et al. (2009) evaluated

the environmental and social benefits of greening around the Kunming urban overpass by hierarchical analysis [7]; Taner R. OZDIL et al. (2015, 2016) evaluated the economic benefits of five landscape projects and two projects by case study method [8,9]. Research on rural regions has been conducted in recent years, and there has also been a shift in the assessment methodology from a single approach to a combination of methodologies. Zhao Yue (2017) used entropy power method and fuzzy comprehensive evaluation method to evaluate the economic benefits of classical landscape with four famous gardens in Suzhou [10]; Tiezheng Zhao et al. (2019) evaluated the environmental benefits of the six small towns in Zhejiang province by using the analytical hierarchical analysis method and the fuzzy comprehensive evaluation method [11]; Liu Zhe et al. (2020) evaluated the social, environmental and economic benefits of Beijing Arctic Temple Park by means of single-factor quantitative model cluster and real-time online evaluation [12]; Miao Yang et al. (2020) used hierarchical analysis and fuzzy comprehensive evaluation to evaluate the social, environmental and economic benefits of urban and small towns in Zhejiang Siming [13]; Zeng Li et al. (2021) used the hierarchical-entropy power method to evaluate the environmental benefits of the ancient salt culture towns in the intersection area of Sichuan, Yunnan and Guizhou [14]; and Lingyan Xiang et al. (2022) evaluated the environmental benefits and economic benefits of the central park landscape in Qingkou Town by using the hierarchical analysis process-collaborative variation weight comprehensive evaluation and analysis method [15]. This paper summarizes and compares the existing landscape performance evaluation methods and summarizes the advantages and disadvantages them. However, there is a lack of sufficient empirical research on the landscape performance evaluation of microlandscapes in rural areas. The evaluation indicators are not clear, and the applicability of the evaluation method has also not been proven, so summarizing a set of universal landscape performance evaluation systems for rural microlandscapes is a crucial step.

Based on the experience and knowledge of the decision-makers, the AHP method makes a more accurate judgment of the connotation and extension of the evaluation index, which reflects the intention of the decision-makers [16], and the final weight is more authoritative. AHP is frequently utilised in multiple-objective decision-making, preference connection analysis, information credibility, renewable energy, and other domains, as a subjective decision-making technique [17–20]. The AHP approach is able to manage the whole assessment index more thoroughly and raise the overall credibility of the evaluation findings when compared to other subjective weighting techniques like the Delphi method and cloud model method [21–23]. However, this method is still susceptible to the decision-makers' subjective thinking, past experience and personal preference, and so the construction weight lacks stability.

The entropy method is an objective empowerment method that determines the weight according to the correlation between indicators and the variation in internal sample data; it avoids the subjective deviation caused by human factors. In recent research, the entropy weight approach has been integrated with other ways to weight evaluation indices in the assessment of flood risk, performance of rail transit operations, urban low-carbon indicators, carrying capacity of resources and the environment, sensors, and industrial robots [24–29]. Entropy weighting technique has an advantage over other objective weighing methods like principal component analysis method and coefficient of variation method in that it can discriminate between different levels of internal data of indicators [30–32]. However, in reality the entropy weight method is significantly impacted by data variation, which frequently leads to the phenomenon that the determined weight is inconsistent with the actual importance degree of the attribute and prevents it from fully assessing the importance degree of various attribute indexes from the overall evaluation.

TOPSIS method uses different dimensions of multiple indicators of unified dimensional processing, while eliminating the influence of different dimensions of the evaluation object comprehensive ranking. In recently published research, this methodology is frequently used to AI computing, material selection, chip communication, human resources, and other topics [33–36]. VIKOR, PromeTHEE-II, and other decision analysis methods

have certain drawbacks, such as constant weight coefficient standard and high subjectivity of dominating function [37,38]. ]; while the TOPSIS approach is more versatile in application and is more compatible with different weighting methods. In practice, however, because the weight of different evaluation indicators is not the same and because of the great subjectivity and blindness [39], the AHP method combined with the entropy method empowers the TOPSIS method to compensate for the single TOPSIS method's insufficiency to empower the evaluation index.

Combination Weighting method combined with TOPSIS method produces an evaluation method that has been widely used in environmental engineering, safety engineering, business administration and other disciplines. For example: Hu Quanguang et al. (2017) used the CW-TOPSIS method to evaluate the impact of rock grade on safety standards [40]; Li Yunyan et al. (2017) combined the hierarchical analysis method and the entropy weight method to develop a comprehensive development index of low-carbon cities in the four municipalities that was evaluated and ranked [41]; Jing-Jing Wang et al. (2018) assigned the weight of the robot processing language information decision criterion by using the combined weighting technique [42]; Tao Peng et al. (2020) used the cloud model and combined weighting method to study the carrying capacity of water resources in Guiyang [43]; and Yu-shan Hu et al. (2021) used the combined empowerment method to evaluate the credit level of 115 road transport enterprises in a certain province in China [44].

This combined empowerment method not only controls the weight of important indicators at the subjective level but also carries information by real reaction data from the objective level. The main purpose of rural microlandscape projects is to improve the living environment, but they hardly produce economic benefits; due to the limitation of microlandscape volume, the limited ability to intercept rainwater and carbon sinks, such projects do not improve the regional microclimate so the environmental benefit is relatively poor. Therefore, the social benefit is an important aspect to evaluate the quality of rural microlandscape projects. The CW-TOPSIS method adapted to the rural microlandscape project gives priority to social benefits and the living environment. Based on the continuous principle, combined with local villagers from actual life and relatively subjective weight, the index weight is not completely affected by data fluctuations and so the use of the entropy method to ensure the objectivity of the quantitative evaluation accurately reflects the number of artificial subjective feelings of different objective effects. Therefore, the CW-TOPSIS method is most applicable to microlandscape projects in rural areas.

Based on the concept of a "sustainable living environment", this study establishes an evaluation system adapted to rural microlandscapes in highly urbanized areas, which includes evaluation indicators and evaluation methods. First, this study refers to the existing research cases of LAF, classifies and summarizes the research cases according to the characteristics of the rural microlandscape. Taking social benefits as an example, extracts the five criteria of rural microlandscape evaluation indicators, and constructs the evaluation index system, including the indicators of social, environmental, and economic benefits. Second, for the construction of the evaluation model, after comprehensively discussing the existing evaluation methods of landscape performance, the TOPSIS evaluation method based on the AHP method + entropy weight method is proposed by referring to the existing evaluation methods of relevant disciplines. Finally, the microlandscape projects of highly urbanized villages in the southeastern coastal areas of China were selected for empirical evidence and analysis. By comparing the actual evaluation of the villagers, the rationality of the evaluation model was calculated and tested, as shown in Figure 2.

The innovations and main contributions of this paper are as follows:

Reference the evaluation index in *Evaluation of Beautiful Rural Construction* and LAF, combining the characteristics of small volume, low cost, high efficiency of rural microlandscapes, and the problems found in design, use and maintenance stages during field research; build a set of appropriate tools for rural microlandscape evaluation index, help to offset the current lack of evaluation indexes for such projects;

Combining the subjective weighing method AHP with the objective weighting technique entropy weighting method creates a combination weighting approach. It can influence the relationship between various assessment indicators on a subjective level and also play to the correctness of objective data, making each evaluation indicator's weight more in line with reality;

The combined weighting method was innovatively introduced to improve the accuracy of the evaluation when establishing the landscape performance evaluation model of rural microlandscapes in highly urbanized areas, giving full play to the advantages of the weighted TOPSIS model, and avoiding the shortcoming of the single TOPSIS method that is too different from the actual ranking result due to the influence of index weight. It provides reference for specific quantitative analysis and accurate feedback to the project;

Based on the analysis of the advantages of the top-ranked microlandscape projects and the disadvantages of the bottom-ranked projects, four strategies are proposed based on the results that stress relief and the number of visitors account for the largest weight in the combined weights: the early design should consider the late maintenance; mobilize the enthusiasm of villagers to participate; cooperate with the village government and universities; and consider the fairness of the use of vulnerable groups. It enhanced the evaluation and feedback mechanism of the whole life cycle of rural microlandscape projects.

Thus, this research is suitable for publication in high-quality journals.

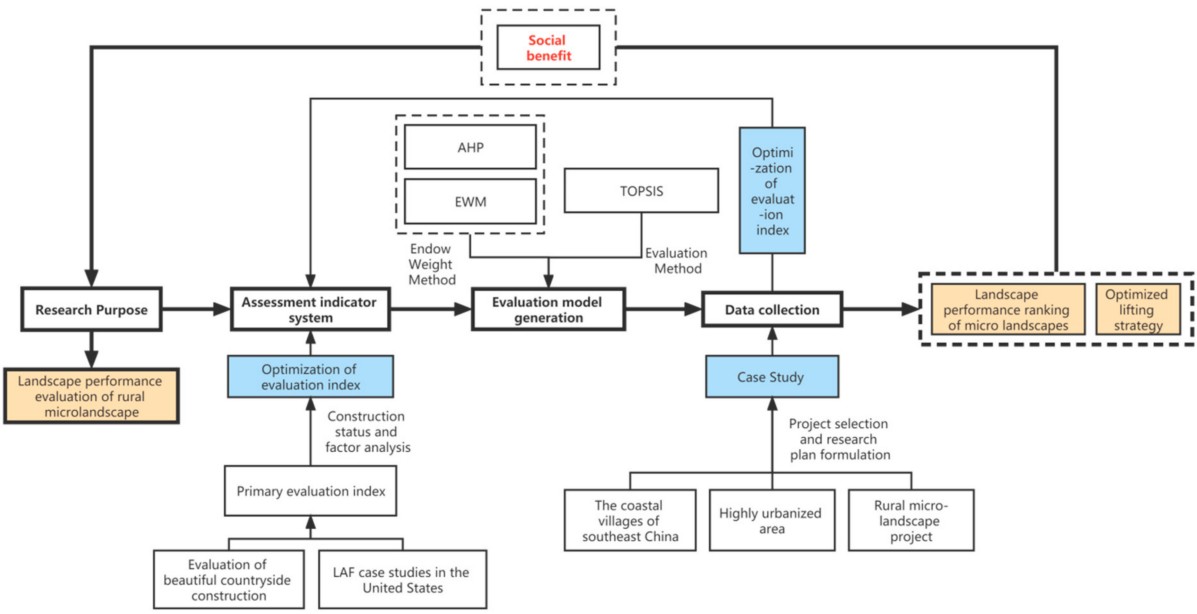

**Figure 2.** Landscape performance evaluation process.

## 2. Materials and Methods

### 2.1. Study Area

This institute used: data from the author and team members on 26 November 2021, ~28 surveys of Quanzhou city, Fujian province Yinglin town, Sanou villages, Huwei village, Dongpu village; 25 microlandscape project sample data; research from the social, environment, economy three levels formulated from the villagers' questionnaire; village committee inquiry table; field observation record and microlandscape VR scene perception table; for a total of 43 related indicators from the data collection. The data objectively reflect the current use of rural microlandscapes in highly urbanized areas represented by various towns in Jinjiang city and provide practical sample data for the construction of a landscape performance evaluation model of rural microlandscapes.

Yinglin town, located in Jinjiang city, Quanzhou city, Fujian Province, currently has a registered population of 47,610 (2020). Over the past decade, its population grew by approximately 6%, or 2910 people (44,700 in 2010). However, the current permanent resident

population of Yinglin Town is 102,070, which is much higher than the registered population; there are many migrants [45,46]. The high proportion of migrants is mainly due to the many enterprises in Yinglin town in the textile and garment manufacturing industry. Since the reforms and opening up agenda, local villagers have taken advantage of the idle funds of overseas Chinese citizens who are from Yinglin Town (and other towns in Jinjiang City) to raise funds to build township enterprises, greatly promoting the rapid economic and social development of Jinjiang City, which is also known as the Jinjiang Model. Therefore, different from towns in other parts of China, Yinglin Town has three characteristics: first, a high degree of urbanization; second, there are many abandoned old houses in the village; and third, the high proportion of migrant population. Rural areas in other parts of China are declining, and some even have hollowed out villages. However, the southeastern coastal areas of China represented by Yinglin town have rarely experienced serious population loss. In contrast, optimizing the construction environment within rural areas and improving the sustainable living environment are the main challenges faced by such rural areas. Since 2017, the local government of Jinjiang City has cooperated with various parties to carry out several rural microlandscape construction practices, attracting college students, professionals, and villagers to participate in rural microlandscape construction, greatly improving the appearance and the rural living environment with limited funds. However, many of the completed microlandscape projects have only considered the beautification of the environment and lack consideration for the public activities of the residents, resulting in a low utilization rate of the microlandscape. In addition, due to the insufficient consideration of the later operation and maintenance in the design stage, it is difficult to maintain the projects in the later stages, and the built microlandscapes are even abandoned, which makes sustainable use difficult. The 25 research cases selected in this study are from Sanou Village, Huwei Village, and Dongpu Village, which are administrative villages under the jurisdiction of Yinglin Town. The basic overview of the research cases is shown in Table 1.

**Table 1.** Overview of the basic information of the study cases.

| Project Number | Aerial Photo | Node Diagram | Area/m$^2$ |
|:---:|:---:|:---:|:---:|
| A1 | | | 210 |
| A2 | | | 421 |
| A3 | | | 159 |

**Table 1.** *Cont.*

| Project Number | Aerial Photo | Node Diagram | Area/m² |
|:---:|:---:|:---:|:---:|
| A4 |  |  | 480 |
| A5 |  |  | 691 |
| A6 |  |  | 627 |
| A7 |  |  | 374 |
| A8 |  |  | 799 |
| B1 |  |  | 664 |

**Table 1.** *Cont.*

| Project Number | Aerial Photo | Node Diagram | Area/m² |
|:---:|:---:|:---:|:---:|
| B2 |  |  | 833 |
| B3 |  |  | 245 |
| B4 |  |  | 115 |
| B5 |  |  | 111 |
| B6 |  |  | 761 |
| B7 |  |  | 304 |
| B8 |  |  | 155 |

**Table 1.** *Cont.*

| Project Number | Aerial Photo | Node Diagram | Area/m² |
|---|---|---|---|
| B9 |  |  | 668 |
| B10 |  |  | 122 |
| B11 |  |  | 98 |
| B12 |  |  | 284 |
| B13 |  |  | 329 |
| B14 |  |  | 343 |
| C1 |  |  | 296 |

**Table 1.** *Cont.*

| Project Number | Aerial Photo | Node Diagram | Area/m$^2$ |
|:---:|:---:|:---:|:---:|
| | 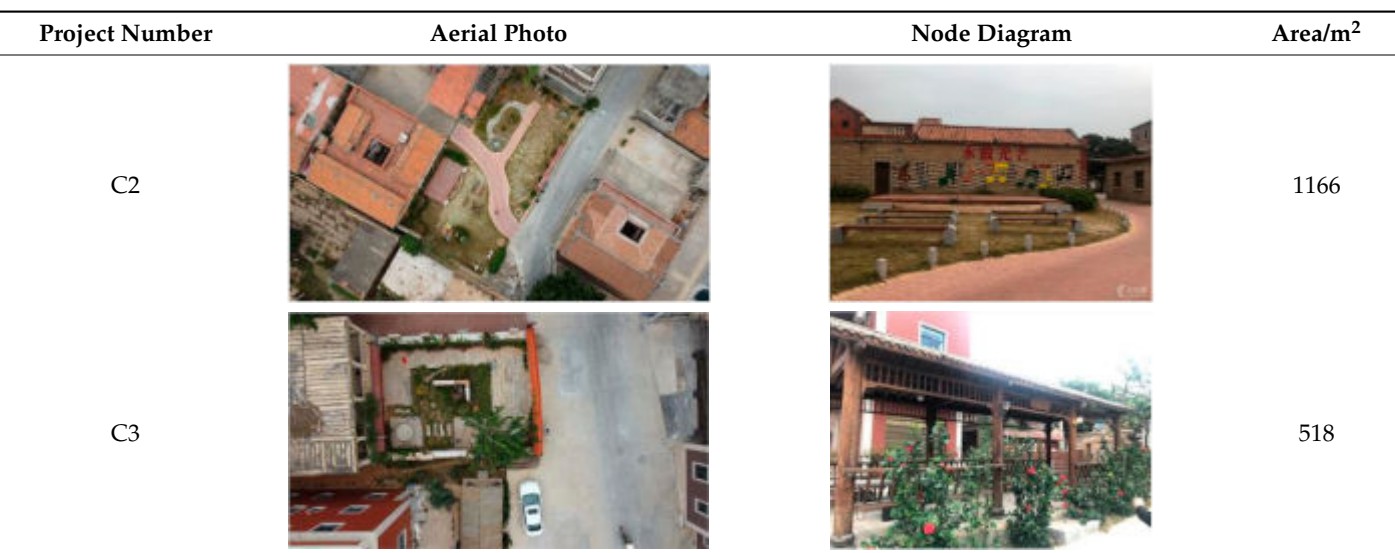 | | |
| C2 | | | 1166 |
| C3 | | | 518 |

### 2.2. Evaluation Indicator System and Data Collection

Based on Maslow's demand theory and referring to the relevant content of social performance evaluation index in LAF landscape performance, this paper evaluated the social performance of rural micro-landscape in Jinjiang from five aspects: "Meet physiological needs", "Meet security needs", "Meet adscription needs", "Meet cognitive needs", and "Meet self-fulfilment needs". Among them, "Meet physiological needs" refers to "Comfort and Health", "Meet security needs" refers to "Safety and Accessibility", "Meet adscription needs" refers to "Social and Service", "Meet cognitive needs" refers to "Beautiful and Education", "Meet self-fulfilment needs" refers to "Culture and Inheritance".

Comfort and Health: The route and physical scale of the site affect the site's comfort. A walkway that is excessively crowded or winding may lessen the site's amenities. The pedestrian environment and open space were taken into consideration when designing Scott Park and Brooklyn Bridge Park [47]. Unrestricted internal paths and the proper spatial size can increase users' comfort and security while using rural microlandscapes. In order to evaluate the social advantages of microlandscapes, "unimpeded internal route" and "suitable spatial size" might be chosen. Eliseo Collazos Fog Water Farm, a small-scale landscape example, demonstrates how space may significantly enhance users' mental health and outdoor activities. As assessment markers for the social advantages of microlandscapes, "reducing emotional tension" and "enhancing outdoor activities" might be used.

Safety and Accessibility: This study revealed that there are many old and young persons who are frequently immobile or have weak physical capabilities. Perfect lighting, barrier-free access, and signage facilities can guarantee safe usage of the microlandscape and increase security. According to pertinent studies, enlarging the visible field and adding illumination might enhance the site's feeling of security [9]. In order to evaluate the social advantages of microlandscapes, "signage facilities," "barrier-free facilities," and "lighting facilities" might be chosen. The term "accessibility" relates to how easily a location may be "approached/reached," which is often measured by distance, time, cost, and the destination's appeal and demand. Users' "transportation mode time" and their "route passage difficulty" to get to the microlandscape both represent the microlandscape's "accessibility", [48,49] hence "transportation mode time" and "route passage difficulty" can be used as assessment indices of the social benefits of the microlandscape.

Social and Service: Research on Erie Street Plaza, Sundance Square Plaza, and other case studies has demonstrated that the appeal of green spaces in parks is mostly due to the improvement of social contact, the availability of a variety of activities, and an increase in visitors. Users have various physical, psychological, and spiritual demands as a

result of their diverse demographic traits, which include differences in their ages, genders, vocations, etc. The best strategy to generate social and service advantages is to provide a lot of locations where people can go and do things, based on the spatial behaviour traits of individuals with diverse features [50]. As a result of staying, "visitors", "long", "public activities richness", "residents access frequency", and "increase the social time" can be chosen as the evaluation index of landscape social benefits "social" dimension; this can build long-term practice and daily life common situations, giving residents a sense of pride and belonging, and enhancing social cohesion [51]. It is possible to use "facing a range of users" and "raising villagers' pride" as assessment indicators of the social benefits of the microlandscape in the "service" component.

Beautiful and Education: An on-the-spot investigation revealed that microlandscape projects will improve the aesthetic quality of the villages. These projects and their surrounding environments, which bring out the best in each other and further improve the comfort of the environment, can be chosen as the landscape social benefits "beautiful" dimension of evaluation indices for their emphasis on plant aspect, hue richness, and harmony with the surroundings. Some microlandscapes used in Jinjiang's microlandscape construction provide local plant education through planning research activities. To evaluate the "education" component of the social benefits of microlandscape, the evaluation index "increasing the understanding of local flora" was chosen.

Culture and Inheritance: The cultural traits of a rural empty space depend on a variety of elements, including the area's nature, surroundings, and established role. The culture that has been profoundly imprinted in an empty area may be explicit and obvious, implicit and thoroughly explored, or both. Explicit visible culture is directly embodied and preserved in the form of objective things, such as traditional structures with regional features, historical artefacts, traditional farming implements, local materials, cultural monuments, and so on left in the area. The artistic value, historical value, cultural value, and other abstract culture that are implicit and must be thoroughly explored. Examples include the traditional construction technology present in the space, historical occurrences, celebrity culture, production, way of life, folklore, opera music, dialect, customs, and collective memory of the villagers. The assessment indices of the social advantages of microlandscapes therefore included "preserving cultural aspects", "using local resources", "using traditional skills", and "enhancing the cognition of traditional culture".

Based on the sustainable theory of the living environment, referring to the *Evaluation of Beautiful Rural Construction* and the LAF landscape performance evaluation index, more appropriate indicators are added, the meaningless index is removed, and a more appropriate qualitative and quantitative evaluation index system is deepened and developed. The social benefits are taken as an example, as shown in Table 2, covering the characteristics of the regional environment.

**Table 2.** Comprehensive evaluation index system of the social benefits of landscape performance.

| Target Layer | The Standard Layer | Index Layer | Processing Method |
|---|---|---|---|
| | Comfort and Health B1 | Degree of internal path patency C11 | Observation and inquiry |
| | | Spatial scale suitability C12 | VR + SD method level 5 scoring |
| | | Relieve emotional stress C13 | VR + SD method level 5 scoring |
| | | Increase outdoor activities C14 | (Approved quantity/Total quantity) × 100% |
| | Safety and Accessibility B2 | Identification facility, C21 | (Quantity/project area) × 100% |
| | | Accessibility facility C22 | (Quantity/project area) × 100% |
| | | Lighting facility C23 | (Quantity/project area) × 100% |
| | | Traffic time: C24 | 0.75 for less than 10 min, 0.5 for 10–25 min, 0.25 for less than 25 min, and 0 for more than 1 h |
| | | Route passage difficulty: C25 | Observation and inquiry |

**Table 2.** *Cont.*

| Target Layer | The Standard Layer | Index Layer | Processing Method |
|---|---|---|---|
| Social Benefit A | Social and Services B3 | Visitors number C31 | Number of single-project visitors within 10 min |
| | | Visitor stay time C32 | 0.25 for less than 30 min, 0.5 for 30–1 h, 0.75 for 1–2 h, and 1 for more than 2 h |
| | | Public activity richness: C33 Resident access frequency C34 Increase the social time C35 | Observation and inquiry |
| | | | 1 for every day, 0.75 for more than three times a week, 0.5 for once a week, 0.25 for once a month, 0 for less than once a month |
| | | For multiple users C36 Rest facilities, C37 | (Number of partners/total number) × 100% |
| | | Fitness facilities C38 | Number of different population groups of people |
| | | Health facilities, C39 | (Quantity/project area) × 100% |
| | Beautiful and Education B4 | Plant seasonal and color richness C41 | (Quantity/project area) × 100% |
| | | Environment coordination C42 | (Quantity/project area) × 100% |
| | | Improve the understanding of native plants C43 Enhance the pride of the villagers C44 | VR + SD method level 5 scoring |
| | Culture and Inheritance B5 | Protecting cultural elements C51 | VR + SD method level 5 scoring |
| | | Using local materials C52 | (Approved quantity/Total quantity) × 100% |
| | | Using the traditional techniques C53 | (Approved quantity/Total quantity) × 100% |
| | | Improve the cognition of traditional culture C54 | (Quantity/project area) × 100% |

Since this study was aimed at rural areas, the educational level of surrounding residents was generally low. To obtain more valid data, questionnaires were administered individually, and the form was completed by the interviewer. In the actual investigation process, the simple questionnaire was too subjective, and the villagers sometimes had difficulty understanding the questions, so the research team members used the form of questionnaire + interview in the later stage. The questionnaire is divided into two parts in the content: the first part collects basic information about the respondents, such as age, gender, and education level; the second part includes questions about their use and improvement in the living environment. Answers were provided using a 5-point Likert scale: very satisfied, satisfied, average, dissatisfied, and very dissatisfied.

*2.3. Combination Weighting Method*

Hierarchical Analysis, or AHP, is taught by the University of Pittsburgh professor T.L.A based on a multi-objective decision analysis methodology proposed by Satty in the 1970s. The principle is to decompose the factors related to decision-making into the target layer, criterion layer, scheme layer, and other layers. Through the calculation and comparison of various factors, the weight of different factors is obtained to provide a reference for decision-makers to choose the optimal scheme [52].

Establish a hierarchical structure: According to the analysis of the problem, the factors contained in the problem determine the association and subordination between each factor. According to the common characteristics of these factors, they are divided into the target layer, standard layer, scheme layer, and other levels.

Establish a pairwise judgment matrix: The judgment matrix represents the comparison of the relative importance of the previous level, and the general form is shown in Table 3.

**Table 3.** The general form of the judgment matrix Reprinted with permission from Ref. [53].

| A | B1 | B2 | . . . | Bn |
|---|---|---|---|---|
| B1 | b11 | b12 | . . . | b1n |
| B2 | b21 | b22 | . . . | b2n |
| . . . | . . . | . . . | . . . | . . . |
| Bn | bn1 | bn2 | . . . | bnn |

The *aij* in the judgment matrix: the nine-point scale method is generally adopted (see Table 4 for definition), which is determined after repeated research based on the data, expert opinions, or the experience of system analysts.

**Table 4.** Nine-point Scale and its Definition Reprinted with permission from Ref. [54].

| Scale aij | Definition |
|---|---|
| 1 | Factor i is equally important as factor j |
| 3 | Factor i is slightly more important than factor j |
| 5 | Factor i is significantly more important than it is for factor j |
| 7 | Factor i is much more important than factor j |
| 9 | Factor i is extremely more important than factor j |
| 2,4,6,8 | The scaling value of the significance of factor i and factor j is somewhere between the two adjacent grades mentioned above |
| The inverse of the scaling value | Counter comparison of factor i and factor j: aji = 1/aij |

Calculate the weights of each element:
The maximum eigenvalue of the judgment matrix $\lambda_{\max}$ is first calculated [54]

$$\lambda_{\max} = \sum_{i=0}^{n} \frac{(AW)_i}{nw_i} \tag{1}$$

$\lambda_{\max}$ is the maximum eigenvalue of the judgment matrix $A$, $(AW)_i$ is the $i$-th element of the product of matrix $A$ and eigenvector $W$ and $w_i$ is the eigenvector for the $i$-th element.
The consistency index *CI* of the matrix was calculated [54]

$$CI = \frac{\lambda_{\max} - n}{n-1} \tag{2}$$

Check the table and calculate the proportion of consistency *CR* of the matrix [54]

$$CR = \frac{CI}{RI} \tag{3}$$

*RI* is the average random consistency index, and the value of RI is used to construct 500 sample matrices by the random method and obtain the maximum eigenvalue $\lambda'$max. The specific values of the *RI* are shown in Table 5.

**Table 5.** The mean random concordance index *RI* Reprinted with permission from Ref. [55].

| n | 1 | 2 | 3 | 4 | 5 | 6 | 7 | 8 | 9 | 10 | 11 | 12 | 13 | 14 | 15 |
|---|---|---|---|---|---|---|---|---|---|---|---|---|---|---|---|
| RI | 0 | 0 | 0.52 | 0.89 | 1.12 | 1.26 | 1.36 | 1.41 | 1.46 | 1.49 | 1.52 | 1.54 | 1.56 | 1.58 | 1.59 |

If *CR* < 0.1, the consistency of the judgment matrix is considered acceptable; otherwise, the judgment matrix will be corrected.
Finally, according to the calculated weight, each index is ranked, and the order of importance of the evaluation index is obtained. The flow chart is shown in Figure 3.

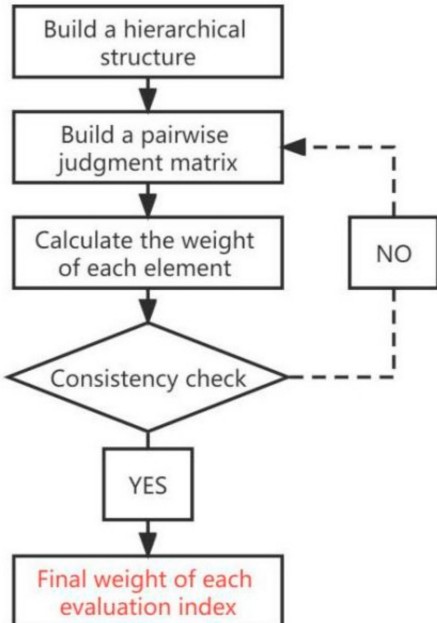

**Figure 3.** Flowchart of the AHP method.

The concept of entropy originally came from thermodynamics in physics, and C.E. Shannon introduced it in 1948, using entropy as a means to measure the amount of information carried by a target object [56]. In information theory, the more information there is, the less uncertainty there is, and the smaller the corresponding entropy is, and vice versa. For different evaluation indices, the greater the difference between the sample data, the greater the overall evaluation results. According to the principle of entropy in information theory, the larger the sample data carries the information, the smaller the entropy; therefore, the corresponding index weight is larger.

Establish the original sample data matrix: Establish the original matrix $X = (x_{ij})m \times n$, and the sample data corresponding to all indicators are forward and standardized. If there was a negative number in the sample data, it was converted to a nonnegative interval [57].

$$z_{ij} = x_{ij} \Bigg/ \sqrt{\sum_{i=1}^{n} x_{ij}^2} \tag{4}$$

$z_{ij}$ is the element corresponding to $x_{ij}$ after nonnegative transformation.

Standardizing matrix $X$ obtains matrix $\widetilde{Z}$, and its standardization formula is [58]:

$$\widetilde{z}_{ij} = \frac{x_{ij} - \min\{x_{1j}, x_{2j}, \cdots x_{nj}\}}{\max\{x_{1j}, x_{2j}, \cdots x_{nj}\} - \min\{x_{1j}, x_{2j}, \cdots x_{nj}\}} \tag{5}$$

Assuming that there are n objects to be evaluated and m evaluation indicators, the nonnegative matrix is obtained after the above treatment [58]:

$$\widetilde{Z} = \begin{bmatrix} \widetilde{z}_{11} & \widetilde{z}_{12} & \cdots & \widetilde{z}_{1m} \\ \widetilde{z}_{21} & \widetilde{z}_{22} & \cdots & \widetilde{z}_{2m} \\ \vdots & \vdots & \ddots & \vdots \\ \widetilde{z}_{n1} & \widetilde{z}_{n2} & \cdots & \widetilde{z}_{nm} \end{bmatrix}$$

Calculate the proportion of each sample for each indicator: Regard the calculated proportion as the probability $p_{ij}$ used in the relative entropy calculation [58]

$$p_{ij} = \frac{\widetilde{z}_{ij}}{\sum\limits_{i=1}^{n} \widetilde{z}_{ij}} \tag{6}$$

It should be guaranteed that the each $p_{ij}$ probability sum corresponding to each indicator is 1.

Calculate the entropy of the index: The information entropy $e_j$ of each index is calculated based on the probability calculated in the previous step [58]:

$$e_j = -\frac{1}{\ln n} \sum\limits_{i=1}^{n} p_{ij} \ln(p_{ij})(j = 1, 2, \cdots, m) \tag{7}$$

where the larger $e_j$ is, the greater the information entropy of the *j*-th indicator is, and the less information it contains. Therefore, the information entropy is forward processed to obtain the information utility value $d_j$ [58].

$$d_j = 1 - e_j \tag{8}$$

It is then normalized to finally obtain the entropy weight $W_j$ for each index [58].

$$W_j = d_j \left/ \sum\limits_{j=1}^{m} d_j \, (j = 1, 2, \cdots, m) \right. \tag{9}$$

The flowchart is shown in Figure 4

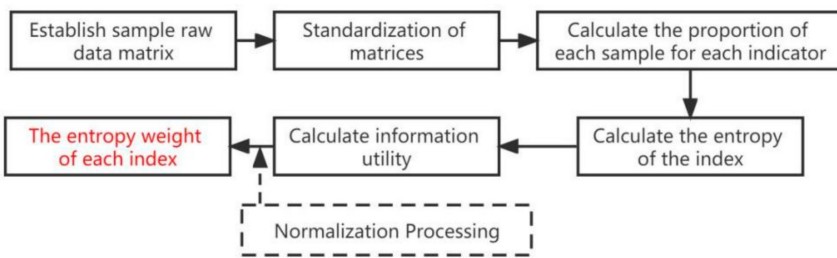

**Figure 4.** Flowchart of the entropy weight method.

*2.4. Ranking Method*

The TOPSIS method is a common comprehensive evaluation method [58,59] that was proposed by Wang and Yoon in 1981 [60]. The TOPSIS method can make full use of the data, and the results comprehensively and systematically reflect the gap between the evaluation index samples. The principle is to calculate the overall maximum and minimum of different evaluation objects by comparing the specific data to compare the distance between different evaluation objects.

Forward conversion of sample data: convert all indicators into very large indicators, that is, the larger the data, the better the benefit

Sample data standardization: The purpose of data standardization is to eliminate the impact of different dimensions so that the sample data of different dimensions can be combined as a whole for processing [61].

$$z_{ij} = x_{ij} \left/ \sqrt{\sum\limits_{i=1}^{n} x_{ij}^{2}} \right. \tag{10}$$

$z_{ij}$ is for each element in the normalized matrix $Z$ and xij is for each element in the unnormalized matrix $X$.

Calculate the score and normalize: select the maximum value $Z_j^+$ corresponding to the evaluation object with the highest score in each evaluation index. The minimum value corresponding to the evaluation object with the lowest score, $Z_j^-$, computes the distance $D_i^+$ between the sample data of the different evaluation objects and them, And $D_i^-$, and ranks after the final results [16].

$$D_i^+ = \sqrt{\sum_{j=1}^{m} \left(Z_j^+ - z_{ij}\right)^2} \tag{11}$$

$$D_i^- = \sqrt{\sum_{j=1}^{m} \left(Z_j^- - z_{ij}\right)^2} \tag{12}$$

Thus, the unnormalized score $S_i$ of the evaluated object was calculated [16]

$$S_i = \frac{D_i^-}{D_i^+ + D_i^-} \tag{13}$$

The resulting $S_i$ normalization was performed [16]

$$\widetilde{S}_i = S_i / \sum_{i=1}^{n} S_i \tag{14}$$

Among them, we should guarantee that $\sum\limits_{i=1}^{n} \widetilde{S}_i = 1$; that is, the corresponding score sum of each evaluation object is 1.

The traditional TOPSIS method can unify multiple dimensionless indicators of different dimensions and obtain a comprehensive ranking of evaluation objects while eliminating the influence of different dimensions. However, in practice, because the weights of different evaluation indicators are not the same, they often need to be given artificially, with great subjectivity and blindness [23].

The introduction of the AHP method provides expert opinion as a reference. It reflects the intention and extension of the decision-makers [16] and the intention of the decision-makers. However, this method is still susceptible to the decision-makers' subjective thinking, experience, and personal preference, and the weight lacks stability.

The entropy method is an objective empowerment method that determines the weight according to the correlation between indicators and the variation in internal sample data and avoids the subjective deviation caused by human factors. However, the entropy method is greatly affected by the data variation.

Given the subjective and objective empowerment methods, some studies will subjectively and objectively use two empowerment methods. The third empowerment method, namely, the Combination Weighting method, combines the advantages of the subjective and objective two empowerment, making up for the shortcomings. The final weight structure has the overall coordination and can objectively reflect the relationship between different indicators.

Finally, the combined weight is calculated to combine the subjective weight w to take into account the intuitive cognition of social benefit factors of landscape performance and the true reflection law of objective survey data with the objective weight web combined. The Lagrangian function is introduced to establish the optimization decision model; and the Euclidean distance function ensures the difference between the main and objective weight and their corresponding preference degree to obtain the ideal CW.

Step 1: Establish an optimization decision model. Let the comprehensive weight be $W_j$ and get CW [62]:

$$\begin{cases} W_j = \alpha w_{Aj} + \beta w_{Bj} \\ \alpha + \beta = 1 \end{cases} \tag{15}$$

In the formula $w_{Aj}$ is the resulting subjective weight calculated for the AHP method and $w_{Bj}$ is the objective weight calculated for the entropy weight method, The main objective preference degree coefficients are

Step 2: Construct the degree of difference consistency between the subjective and objective weights and the corresponding preference coefficient. Introduce the Euclidean distance function $D(w_{Aj}, w_{Bj})$, Establish the equation for the degree of difference between the main and objective weights and the corresponding preference coefficient [62]:

$$\begin{cases} D(w_{Aj}, w_{Bj}) = \sqrt{\sum_{j=1}^{n} (w_{Aj} - w_{Bj})^2} \\ D(w_{Aj}, w_{Bj})^2 = (\alpha - \beta)^2 \end{cases} \tag{16}$$

Combining Formula (15) and Formula (16) yields the ideal comprehensive weight $W_j$.

The final $W_j$ brings in the TOPSIS method and weights each indicator separately to get the final item ranking. Build processes such as Figure 5 shown.

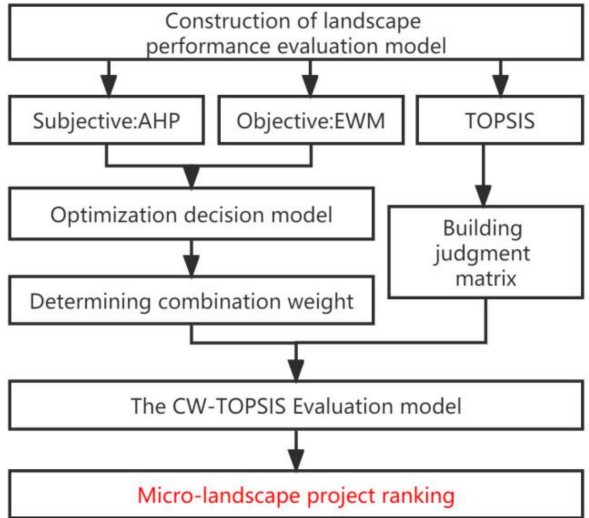

**Figure 5.** Construction process of the landscape performance evaluation model.

*2.5. Evaluation of Landscape Performance Using CW-TOPSIS*

Using the CW-TOPSIS method and taking social benefits as an example, the 25 microlandscape projects in Yinglin Town, Jinjiang City, were ranked. First, invited eight specialists in relevant subjects, including landscape architecture, architecture, environmental engineering, sociology, and others. Of them, four are experts in landscape architecture, two in architecture, one in environmental engineering, and one in sociology. The AHP method was used to give the subjective weight of each specific index, and the CR of the consistency ratio was less than 0.1, which shows that the judgment matrix has satisfactory consistency (Figure 6).

Subsequently, the entropy weight method was used to give the objective weight of each specific index (Figure 7).

The combined weights of the subjective and objective are calculated by the simultaneous Formulas (15) and (16), the weight of the AHP method is 0.6, and the weight of the entropy weight method is 0.4., and the final results are shown in Figure 8.

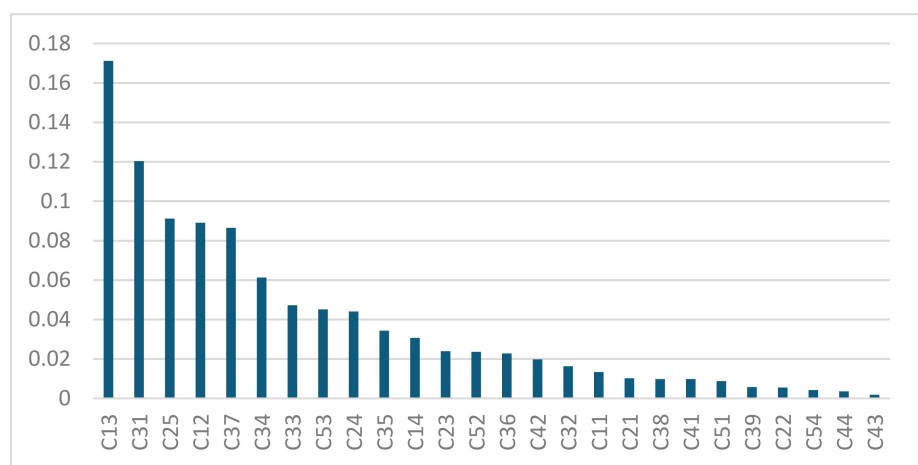

**Figure 6.** AHP method weight ranking results.

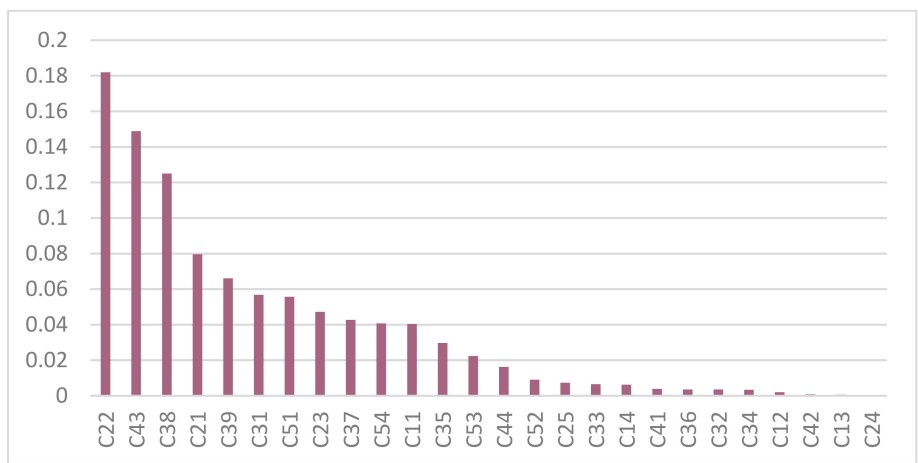

**Figure 7.** Entropy weight method weight ranking results.

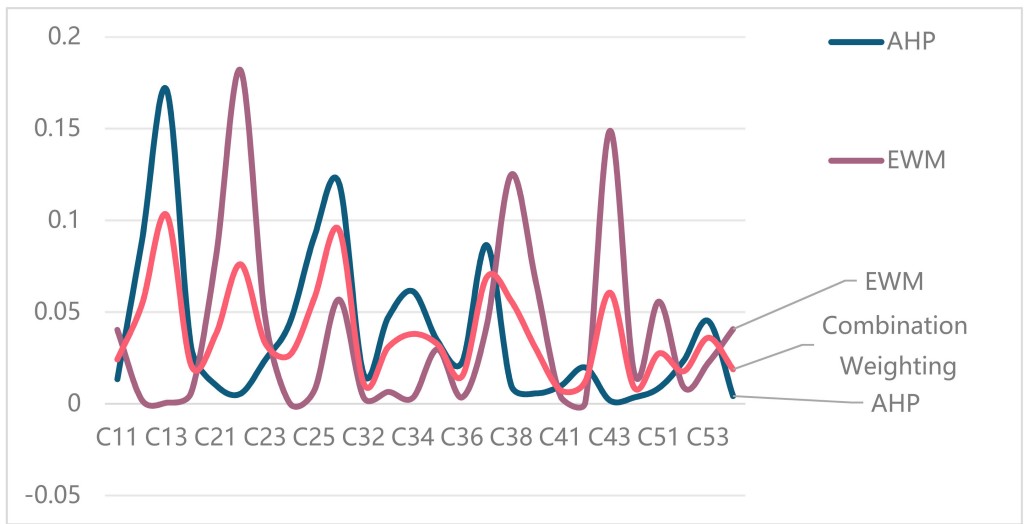

**Figure 8.** AHP, Entropy, and combined assignment.

The results show that the combination weighting eliminates the large weight difference between different evaluation indices when the AHP and entropy weight method are assigned separately and makes the weight of each index more accurately reflect the objective situation. The results of the combination weight ranking for each index layer are shown in Figure 9.

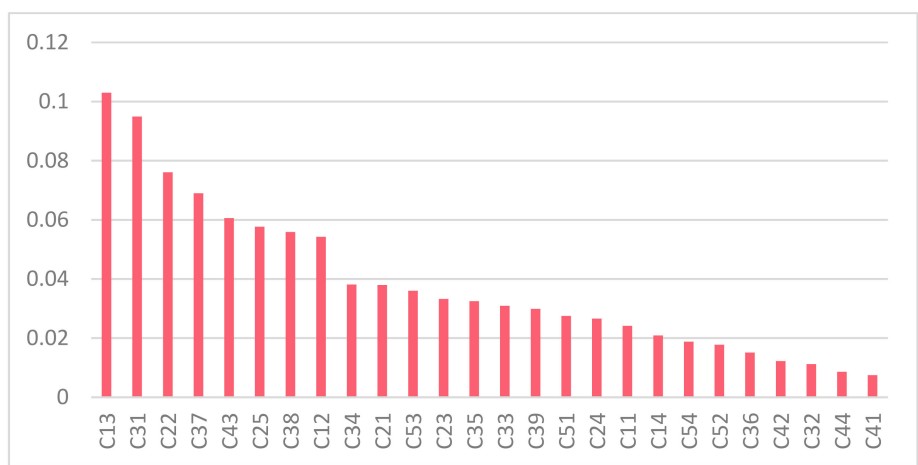

**Figure 9.** Ranking of index layer combination weights.

According to the calculation results of the combination weight, index C13 has the highest weight to relieve emotional pressure and has the greatest impact on social benefits; C41 for plant season and color richness, and C44 has the lowest weight and the least impact. Empowering the resulting combination weights to the sample data yielded the performance ranking of 25 microlandscape projects in terms of social benefits, and the final results are shown in Table 6.

**Table 6.** Social benefit ranking results.

| Item | Positive Ideal Solution Distance D + | Negative Ideal Solution Distance D − | Relative Proximity C | Sorting Results |
|------|------|------|------|------|
| A1 | 0.36 | 0.306 | 0.459 | 1 |
| A8 | 0.415 | 0.253 | 0.379 | 2 |
| A6 | 0.375 | 0.195 | 0.341 | 3 |
| B5 | 0.434 | 0.218 | 0.334 | 4 |
| A3 | 0.41 | 0.199 | 0.327 | 5 |
| C3 | 0.422 | 0.187 | 0.307 | 6 |
| A4 | 0.429 | 0.181 | 0.297 | 7 |
| B3 | 0.429 | 0.177 | 0.292 | 8 |
| B13 | 0.418 | 0.163 | 0.281 | 9 |
| C1 | 0.449 | 0.168 | 0.272 | 10 |
| B4 | 0.445 | 0.165 | 0.271 | 11 |
| A7 | 0.463 | 0.151 | 0.245 | 12 |
| C2 | 0.422 | 0.133 | 0.239 | 13 |
| B10 | 0.443 | 0.129 | 0.226 | 14 |
| A5 | 0.441 | 0.125 | 0.222 | 15 |
| A2 | 0.455 | 0.129 | 0.22 | 16 |
| B7 | 0.428 | 0.118 | 0.217 | 17 |
| B14 | 0.458 | 0.107 | 0.189 | 18 |
| B9 | 0.469 | 0.104 | 0.181 | 19 |
| B2 | 0.46 | 0.1 | 0.179 | 20 |
| B1 | 0.47 | 0.098 | 0.173 | 21 |
| B8 | 0.469 | 0.081 | 0.147 | 22 |
| B11 | 0.478 | 0.075 | 0.135 | 23 |
| B6 | 0.481 | 0.074 | 0.133 | 24 |
| B12 | 0.49 | 0.049 | 0.091 | 25 |

## 3. Result and Discussion

### 3.1. Analysis of Combination Weighting Results

The top five indexes that have the most influence on the evaluation of the performance of the microlandscape in rural regions are: "Relieve emotional stress", "Visitors number",

"Accessibility facility", "Rest facilities", and "Improve the understanding of native plants", as can be seen from the ranking results of the combination weights. "Relieve emotional stress" accounted for the largest proportion of weight, indicating that the microlandscape project has a good healing effect, and also meets the requirements of rural microlandscape construction to improve the quality of living environment. "Visitors number" and "Rest facilities" are all indicators in "Social and Services", indicating that villagers usually do not use microlandscape projects with a strong purpose, but to meet their needs for social activities, choose to go to places with many people, and microlandscapes just provide corresponding activity places. Various rest facilities in the microlandscapes are for villagers to use, which can increase the stay time of the villagers and attract more of them to stop. Therefore, "Social and Services" plays an important role in the social benefits of microlandscape. Among the indicators with the lowest combined weight, three belong to "Beautiful and Education". On the one hand, the villagers' education level is generally not high, and they have no special demand for aesthetics in daily life. On the other hand, the service radius of microlandscape projects can only radiate to the scope within the village, mainly serving the daily rest and activities of villagers, rather than being a landmark place in the village, so villagers pay less attention to microlandscapes in their daily life. "For multiple users" is also relatively low in the ranking of combined weight. In interviews with villagers, many of them say that the purpose of using microlandscapes is to chat with others, while few of them carry out fitness and entertainment activities. Therefore, "Social and Services" and "Beautiful and Education" are not the main focus to improve the social benefits of microlandscape.

According to the weight ranking of AHP method, 1 and 2 are also the top two factors, indicating that there is a consensus on the role of microlandscape in improving the quality of human settlement environment, but 1 is the fourth from the bottom of the results obtained by the entropy weight method. This is because the interview results with villagers show that most microlandscapes can relieve pressure, that is to say, microlandscapes have done quite well in this respect, but the space for optimization and improvement is relatively small. If we want to analyze the problems existing in current microlandscape projects and put forward suggestions for future construction, stress relief is not an indicator that needs to be improved. From the perspective of the weight ranking of the entropy weight method, the most important is 22, but in the result of the AHP method, the weight of 22 only ranks the penultimate. The reason for this situation is that most microlandscape projects do not take barrier-free factors as the key consideration in the early design and actual construction process, which leads to the high score given to the microlandscape projects considering barrier-free facilities design. In general designer and rural government knowledge management, a microlandscape project is low cost, high efficiency to improve the living environment and is convenient, but it ignores the talent of the people who will use the microlandscape, especially the rural older adults and children, in the design of barrier-free facilities that will affect people in the actual use of the convenience.

### 3.2. Analysis of Ranking Results

According to the final ranking results, project A1 is located at the core of the village space, with a large flow of people and high accessibility, and is located in front and the back of the villagers' houses, with a high utilization rate; it has also improved the quality of the villagers' living environment. In addition, in the questionnaire survey on whether the villagers were satisfied with the different microlandscape projects, project A1 also achieved a very high level of satisfaction. The projects closer to the villagers' houses result in the nearby villagers spontaneously organizing the cleaning and maintenance, extending the life of the microlandscape, and making the current situation more effective for serving as the public space in the village for a longer time, as shown in Figure 10.

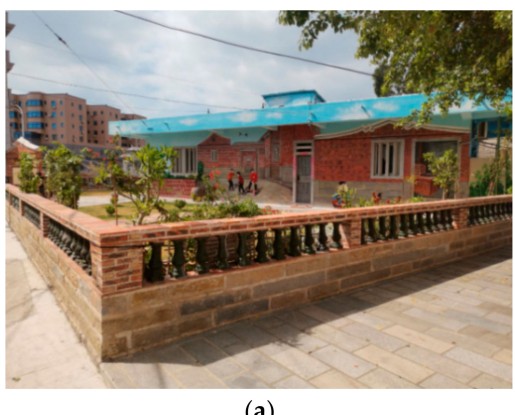
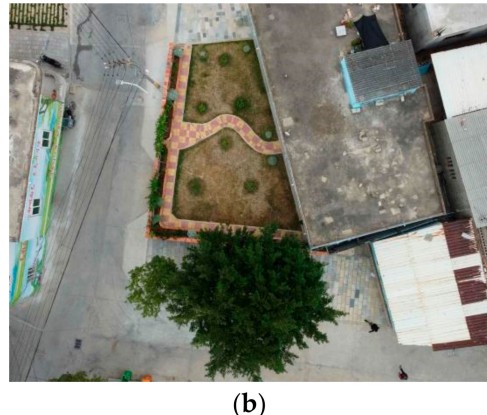

(**a**)                (**b**)

**Figure 10.** A1 node map (**a**) and aerial map (**b**) of the microlandscape project.

The lowest ranking of project B12 is because the south side of the project is a road with many vehicles, which is unsafe and noisy. Secondly, B12 is located in the outskirts of Huwei Village, and except for one or two nearby residents, almost no other residents use it. Finally, on the north side of the project, there are nearby residents who use the abandoned houses to transform the duck house. Due to neglect of care, there is a strong smell near the duck house, which is also one of the reasons why the landscape is rarely used, as shown in Figure 11.

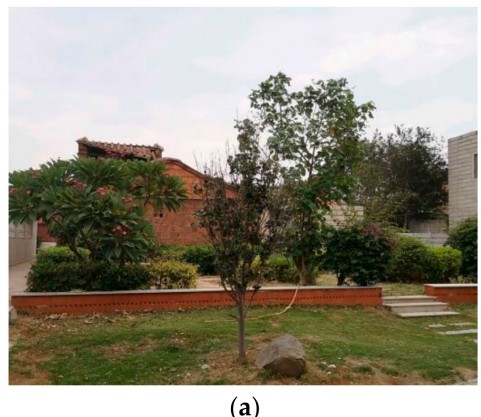
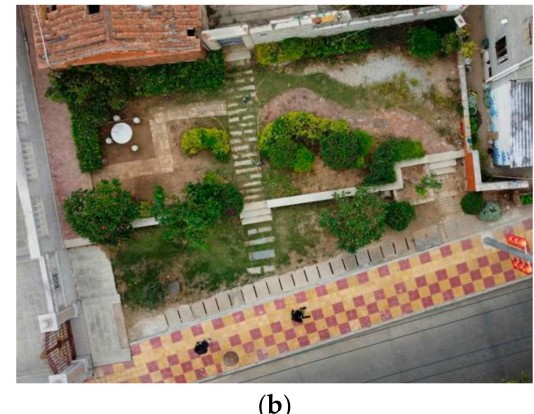

(**a**)                (**b**)

**Figure 11.** B12 node map (**a**) and aerial map (**b**) of the microlandscape project.

*3.3. Discussion*

In conclusion, the evaluation model combining the AHP method and entropy right TOPSIS method adopted in this study accurately and objectively evaluates the ranking of the environmental benefits of rural microlandscape projects. The use of combined weight method avoids the influence of subjective or objective weight method alone on weight. It not only considers the indicators commonly recognized by experts, but also considers the indicators easily ignored, but should not be missing in practical application. The top-ranked projects are generally located in the center of the village, which is conducive to the access and use of villagers. Moreover, the close distance between projects creates a form of cluster distribution, which strengthens the connection between each other and improves the utilization rate. The microlandscapes at the bottom of the ranking are generally located at the edge of the village, close to roads with motor vehicles, with noisy environments and unfavorable access. Because of their remote location, these microlandscapes often exist independently, which makes villagers unwilling to visit and they have a low utilization rate. Therefore, the site selection of the microlandscape project should be arranged in clusters in the village center and sub-center, as close as possible to the residences of the villagers.

This study provides a new idea for the construction of evaluation models in the related landscape performance evaluation system in the future.

## 4. Conclusions and Recommendations

To make the microlandscape a more dynamic open space, designs should eliminate and avoid the use of barriers, such as unsafe lighting and sanitation facilities. To meet the needs of different user groups and use times, the microlandscape design should be inclusive and equally consider the use preferences of local villagers, villagers from neighboring villages, and foreign tourists. Therefore, local specific environmental and sociocultural conditions, environmental changes, and climate adaptation must be considered in the microlandscape design process.

The investigation and analysis of the microlandscape of Yinglin town found that although the highly urbanized rural areas have relatively perfect infrastructure, there are still few public places for residents' activities and few entertainment facilities. With the development of the economy and society and the continuous promotion of rural urbanization, rural residents' requirements for a living environment will continue to improve. In the southeastern coastal areas of China, which have completed urbanization, residents are getting younger and younger. With the introduction of China's three-child policy, young couples with children will account for a large part of the use of the microlandscape. The construction and use of microlandscapes helps to promote rural humanization and beautify the living environment; on the other hand, they help to maintain the ecological balance, improve the health of residents, develop a sense of belonging and improve the happiness index.

This study found that residents' evaluation of the microlandscape depends on many factors, such as functional facilities, culture, education, maintenance, and management. First, there is a common problem of the low utilization rate of rural microlandscapes. The construction of functional facilities can quickly improve the utilization rate of resources and the redistribution of land resources, improve public satisfaction, and form a virtuous cycle of public participation mechanisms. Rural microlandscape projects have the advantages of low cost, fast construction, and quick effect. However, the investigation found that many did not consider maintenance problems in the later stages of the design, resulting in many dilapidated projects and greatly reducing their life. Second, as an important part of the rural public space, the microlandscape needs to inherit and develop the regional cultural characteristics of the countryside. While enjoying green ecology and increasing cultural attributes, local cultural activities can be appropriately introduced, and cultural exhibitions can be held to enhance the younger generation's understanding of local culture and to promote the spread of traditional culture. In addition, sports elements should be appropriately introduced into town parks, and some functional areas (such as badminton courts and basketball courts) can start to run in tidal mode, be paid during the game, at noon or night during the low traffic-free mode, more easily increase public participation, and increase the vitality of rural public space. Through reasonable planning, the income generated by the operations will continue to be invested in the ongoing maintenance of the microlandscapes and their environments, construction and development will be introduced into the ecological development mode. The results of this study provide evidence for the natural environmental inequalities faced by rural Chinese residents, but these inequalities can be reduced by promoting and optimizing rural human settlements in the context of sustainable human settlements.

The following tactics are suggested in an effort to address the current issues with rural microlandscape: ① In the early design stage, later maintenance is considered; but when the design stage lacks later operation and maintenance, this will result in later maintenance difficulties. Some microlandscape maintenance even requires a large amount of village money, and the built microlandscape may even be abandoned, which is difficult to use sustainably. In particular, ordinary villagers and village cadres have not experienced professional design knowledge training, and the operation and maintenance of the living

environment in the early design stage have no concept, leading to various problems in late operations. The microlandscape should include ongoing maintenance in the design stage as much as possible, which is more conducive to later maintenance management strategy. ② Mobilizing the enthusiasm of the villagers to participate in the construction of rural landscape initiatives does not simply refer to a few projects, but to the concept of the process of sustainability of the rural living environment. The designer's responsibility is not only to design one or more microlandscapes, but also to guide the villagers to actively participate in the improvement of the village appearance and spontaneously improve the quality of rural living. At the same time, it is not convenient for rural areas to hire foreign construction personnel to carry out project construction. Moreover, most rural dwellers in China live in settlements linked by blood and clan, and villagers are relatively familiar with each other, so it is convenient for villagers to participate in construction and ongoing maintenance. Microlandscape construction technology requirements are relatively low and easy for villagers to master. ③ The village government and universities cooperate. The lack of estimation of the difficulty of rural construction by designers and evaluation experts leads to difficulty in the implementation of some microlandscape designs. Finally, they are improved by local craftsmen through their own experience. In the process of joint design carried out by the village government combined with colleges and universities, those institutions should be responsible for the achievements of students in the design stage, and university teachers should instill the concept of low technology and light operation and maintenance when cultivating students. For village government cadres, we should also strengthen knowledge in the relevant fields. ④ Considering the fair use by vulnerable groups, it was found that in the survey many microlandscape projects did not take into account the children and older adults, and various differences caused potential safety risks. Care for vulnerable groups such as children and the elderly is not limited to accessibility design, but should also combine the needs of vulnerable groups to create the space needed for their activities. For example, children should receive more dynamic exploratory spaces that provide interest, consider the safety of children's activities, and contribute to their healthy growth; for elderly individuals, they should create more convenient passages and rest spaces for communication.

However, due to the pressures of time and budgets, the measures used in the construction of this evaluation model are all horizontal comparative analyses between different projects, so there is still a large amount of relevant data and analysis in the longitudinal dimension of the same project. Long-term longitudinal comparative analysis of quantitative landscape benefits is indispensable. Second, because this experiment is an evaluation of the completed project, it was difficult to obtain the relevant data before the completion of the project; the construction of the whole evaluation system and the collection of relevant data should be started before the project construction so that the evaluation can more comprehensively reflect the objective benefits of the project. Finally, the survey took place during the policies of prevention and control for the pandemic; the coastal wind was also strong on the survey day, which affected the flow of people, resulting in a slight deviation of the experimental data from the real daily situation. The current study sought to demonstrate the extent that the rural landscape improves the rural living environment; thus, we also need to obtain data analysis to measure the social, environmental, and economic benefits, to determine whether there is adverse mutual influence, and to make the microlandscape projects better able to promote the health and well-being of rural residents. Moreover, the evaluation model constructed in this paper only demonstrates the microlandscape of Yinglin Town, Jinjiang City, Quanzhou, Fujian Province and still needs further demonstration of whether the microlandscape evaluation is generally applicable to other rural areas in the southeastern coastal areas of China. In future work we will further analyze and select the typical villages in other areas to evaluate their microlandscapes, further optimize the evaluation indicators and evaluation methods according to the evaluation results, and improve the rural microlandscapes evaluation system.

**Author Contributions:** All authors contributed to the study conception and design. Concept and the method optimization was completed by L.S. and S.L. Material preparation, data collection and analysis were performed by L.S. and Y.Z. The first draft of the manuscript was written by Y.Z. and M.Y. And all authors commented on previous versions of the manuscript. All authors have read and agreed to the published version of the manuscript.

**Funding:** This study was supported by the Science and Technology Guided Projects of Fujian Province, China (No: 2022N0013).

**Institutional Review Board Statement:** Not applicable.

**Informed Consent Statement:** Not applicable.

**Data Availability Statement:** The study did not report any data.

**Acknowledgments:** This study was funded by Xiamen Key Laboratory of Ecological Building Construction and The Southeast Coastal Ecological Environment, Key Laboratory of Fujian Province Universities.

**Conflicts of Interest:** The authors declare no conflict of interest.

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
