# Peer review of "Combination Weighting Integrated with TOPSIS for Landscape Performance Evaluation: A Case Study of Microlandscape from Rural Areas in Southeast China"

_sustainability, doi:10.3390/su14159794_

Round 1

Reviewer 1 Report

Dear Authors,

The manuscript entitled “Landscape Performance Evaluation of Rural Microlandscapes in Southeast Coastal Towns of China Based on Combination Weighting” I think the manuscript is interesting in the dimension of use in the landscape management of rural areas. However, the presentation of this study seems to be an academic/research report rather than a research article. I have to suggest resubmission only after a thoroughly important issues revision before considering.

1.      The manuscript is too long. Overall, the main drawback of the manuscript is the lack of a scientific form. The text should explain data, methods, and results in a much more focused manner.

I recommend restructuring the manuscript to tighten it. The Introduction part, the Research review part, and some parts of the Research methods should be merged.

2.       The authors should specify/point clearly to the novel of this research to be suitable for publication in high-quality journals.

3.       Based on the Abstract, many beneficial information is missing, such as the significance of this study, research findings, and the knowledge gap.

4.      Introduction part, the reasons for selecting this topic and using these villages as study cases have been explained quite clearly; however, why choose the research tools in the study, there is still no clear specification. What scientific question(s) can be solved via this study is totally unclear. What knowledge gap(s) from the literature review can be summarized is still unclear. In other words, whether theoretical or practical or both contributions, as well as the main novelty of this study, are unclear.

5.      The authors stated that this research aimed to build a more scientific and quantitative landscape performance evaluation model that combines subjective and objective aspects by combining the AHP method commonly used in LAF with the TOPSIS method of entropy through the landscape performance evaluation proposed by LAF and ranks the evaluation objects by using the solution distance method. However, it seems that the results are not precise into the response to this objective

6.      Discussion part, the results finding should be discussed in terms of spatial dimension, tools/methods, factors, and limitations.

7.      I recommend some references for citation that will benefit your research manuscript for restructurings of the manuscript, methodology, and discussion part.

Rubayet Karim and C. L Karmaker. Machine Selection by AHP and TOPSIS Methods. American Journal of Industrial Engineering. 2016; 4(1):7-13. doi: 10.12691/ajie-4-1-2

Wijitkosum, S.; Sriburi, T. Fuzzy AHP Integrated with GIS Analyses for Drought Risk Assessment: A Case Study from Upper Phetchaburi River Basin, Thailand. Water 201911, 939. https://doi.org/10.3390/w11050939.

8.      How many experts do you choose for the AHP process?

9.      There is an Error! Reference source not found., in many parts.

10.  Data presentations in some Tables are confusing. I recommend improving the presentation of many Tables, such as Table 1, and Table 6.

11.  The authors should specify the data source in the Table, such as Table 3 and Table 4. Similarly, all equations should be referenced.

12.  Please check the reference style and follow the journal instructions.

Author Response

Dear Reviewer:

  Thank you provide me with this precious opportunity to modify the article, I have according to your Suggestions, and combined with other advice from the reviewers, and modify my article carried on the comprehensive, hope this change can make my essay achieve the requirement of the journal, and I hope that with your help, the article can see publication at an early date.

Best wishes

Reviewer 2 Report

Dear Authors,

I found your work very interesting. The idea set behind the text meets the scope of the Journal as the article refers to sustainable urban design in terms of optimalization of the landscape performance of rural microlandscapes. 

In general, I accept the text in the present form. I would like to pay your attention to minor language and editorial remarks:

L39-40 – please reconsider the use of term elderly and instead use of the term ‘older adults’

https://journals.lww.com/jgpt/fulltext/2011/10000/use_of_the_term__elderly_.1.aspx

L 55 – please use singular form ‘a mean’

L 65-66; 92-93; 313; 333; 337; 338; 370; 502; 505; 628; 639; 649 – please update the references in the text 

L 105 – please put ‘.’ before the sentence ‘Of the 170percpent (…)’.

L 555 – the word ‘design’ lacks ‘n’.

Best regards,

The reviewer

Author Response

(The authors gave the same response as above.)

Round 2

Reviewer 1 Report

Dear Authors,

Thank you for revise manuscript following the suggestions.